# B_2_O_3_-Doped LATP Glass-Ceramics Studied by X-ray Diffractometry and MAS NMR Spectroscopy Methods

**DOI:** 10.3390/nano11020390

**Published:** 2021-02-03

**Authors:** Wioleta Ślubowska, Lionel Montagne, Olivier Lafon, François Méar, Konrad Kwatek

**Affiliations:** 1Faculty of Physics, Warsaw University of Technology, 00-662 Warsaw, Poland; konrad.kwatek@pw.edu.pl; 2UMR 8181-UCCS-Unité de Catalyse et Chimie du Solide, University Lille, CNRS, Centrale Lille, University Artois, F-59000 Lille, France; lionel.montagne@univ-lille.fr (L.M.); olivier.lafon@univ-lille.fr (O.L.); francois.mear@univ-lille.fr (F.M.); 3Institut Universitaire de France, 75005 Paris, France

**Keywords:** all-solid-state batteries, solid electrolyte, ionic conductors, lithium-ion conductors, lithium aluminum titanium phosphate (LATP), glass-ceramics, nuclear magnetic resonance (NMR) spectroscopy

## Abstract

Two families of glasses in the Li_2_O-Al_2_O_3_-B_2_O_3_-TiO_2_-P_2_O_5_ system were prepared via two different synthesis routes: melt-quenching and ball-milling. Subsequently, they were submitted to crystallization and yielded the Li_1.3_A_l0_._3_Ti_1.7_(PO_4_)_3_ (LATP)-based glass-ceramics. Glasses and corresponding glass-ceramics were studied by complementary X-ray diffraction (XRD) and ^27^Al, ^31^P, ^7^Li, ^11^B magic-angle spinning nuclear magnetic resonance (MAS NMR) methods in order to compare their structure and phase composition and elucidate the impact of boron additive on their glass-forming properties and crystallization process. XRD studies show that the addition of B_2_O_3_ improves the glass-forming properties of glasses prepared by either method and inhibits the precipitation of unwanted phases during heat treatment. MAS NMR studies allowed us to distinguish two LATP phases of slightly different chemical composition suggesting that LATP grains might not be homogeneous. In conclusion, the crystallization of boron-incorporated LATP glasses can is an effective way of obtaining LATP-based solid state electrolytes for the next generation of lithium-ion batteries provided the proper heat-treatment conditions are chosen.

## 1. Introduction

For the last few decades, lithium (Li)-ion batteries (LIBs) have become the main source of energy for a variety of portable electronics as well as for hybrid and electric vehicles. Although Li-ion batteries are very efficient in terms of energy density, they suffer from limited cyclability and safety issues. Therefore, developing a new generation of LIBs seems crucial for satisfying the growing demand for safer, more efficient and reliable electrochemical cells [1,2,3]. In this context, all-solid state lithium-ion batteries (ASSBs) have attracted great interest. Replacing liquid electrolytes by its solid counterparts offer many advantages, e.g.,: improved safety, elimination of toxic leakages, non-volatility, low flammability, thermal and mechanical stability, reduced self-discharge, higher power density as well as increased cyclability [4,5,6].

As the choice of an appropriate solid electrolyte is a key factor ensuring the good performance and durability of ASSBs [7,8,9], various types of solid Li^+^ conductor have already been studied, including oxide and sulfide compounds, such as perovskite-type Li_3x_La_(2/3-x)_TiO_3_ (LLTO) [10,11,12], garnet-type Li_7_La_3_Zr_2_O_12_ (LLZO) [13,14], NASICON-structure phosphates LiTi_2_(PO_4_)_3_ [15,16,17] or LISICON-type compounds [18,19,20].

One of the most promising candidates for application in ASSBs is lithium aluminum titanium phosphate with a chemical formula Li_1.3_Al_0_._3_Ti_1.7_(PO_4_)_3_ (LATP). This compound is derived from LiTi_2_(PO_4_)_3_ system of NASICON-type structure [21,22] where partial substitution of Ti^4+^ by Al^3+^ takes place [23]. It crystallizes with the R$\bar{3}$c space group of rhombohedral symmetry. LATP has been recognized for its high thermal stability, non-flammability, favorable mechanical properties, stability against water and remarkably high bulk (grain interior) ionic conductivity which can reach up to 10^−3^ S/cm at 300 K [24]. Moreover, this compound can be obtained more simply (using processing temperature lower than 900 °C) and with higher reproducibility than other solid state electrolytes.

LATP ceramics prepared by the sintering of polycrystalline powders have considerably lower total ionic conductivity ranging from 10^−5^ to 10^−4^ S/cm depending on the synthesis method [25,26,27,28,29]. One reason for that significant decrease in total conductivity with respect to the bulk one is the precipitation of secondary non-conducting phases at grain boundaries (e.g., TiP_2_O_7_, TiO_2_, AlPO_4_) which, if present in high concentrations, block pathways for Li^+^ ions between the highly conductive grains [30]. The other factors impeding the conduction in LATP ceramic are related to microstructure: high porosity or presence of microcracks which may also hinder the ionic transport through grain boundaries [31,32].

One of the most promising preparation procedures which is believed to address some of the aforementioned problems is the glass-ceramic route. This approach consists of preparing a LATP-based glass and submitting it to a heat treatment in controllable, precisely adjusted conditions to allow the precipitation of LATP phase (nucleation) and the subsequent grain growth. When compared to their sintered polycrystalline analogues, glass-ceramics usually have denser microstructure with reduced porosity and grain boundary effects, hence they are characterized by better mechanical and electrical properties [33].

The glasses in the Li_2_O-Al_2_O_3_-TiO_2_-P_2_O_5_ system can be prepared by a conventional melt-quenching method in which a mixture of reagents is melted at high temperature (usually 1400–1500 °C), then poured and rapidly “quenched” between two stainless steel plates to a configurationally frozen, glassy state of a desired shape and size. This glass is used as a precursor to obtain the glass-ceramic material of a given chemical composition and/or microstructure by an appropriate heat treatment.

By crystallizing the LATP precursor glasses, highly conducting glass-ceramics with the maximum conductivity of 10^−4^–10^−3^ S/cm at room temperature were obtained [34,35,36,37]. Despite its success, a number of issues concerning the glass-ceramic route still remain unresolved. First of all, the melting of reagents requires both high temperatures (~1400 °C) and long processing time (1–1.5 h), which is believed to cause lithium loss at T > 1000 °C. Moreover, LATP glasses have rather poor glass-forming properties and require fast cooling rates. Otherwise, they immediately crystallize. Secondly, during heat-treatment not only the main LATP phase precipitates but also some secondary, poorly conducting phases, e.g., AlPO_4_, TiP_2_O_7_ and LiTiPO_5_, may be present at grain boundaries and decrease the total conductivity.

One of possible ways to modify the properties of the Li_2_O-Al_2_O_3_-TiO_2_-P_2_O_5_ glasses is to use appropriate additives to the melt. Boron trioxide, B_2_O_3_ has been used not only as an additive improving the glass-forming properties of the melt but also as an agent influencing the crystallization kinetics and impeding the precipitation of secondary, unwanted phases [38,39,40,41]. In our previous study [42] we investigated this effect and showed that the ionic conductivity of LATP-based glass-ceramics can be increased due to B_2_O_3_ addition. If the parent LATP glass is prepared by melt quenching, the high melting temperatures lead to the lithium loss. Therefore, there is a strong need to develop low-temperature preparation methods, such as the amorphization by means of mechanical milling (ball-milling technique). Mechanical milling is a simple, inexpensive but powerful processing technique which allows the stabilization of non-equilibrium phases, namely the production of amorphous powders starting from blended mixtures of reagents [43,44,45]. Highly Li^+^ conductive LATP glass-ceramics have been successfully prepared from amorphous powders by mechanical milling [46] and exhibited the total ionic conductivity of as high as 10^−3^ S/cm at room temperature [47].

This work is a comparative study of two families of LATP-based glass-ceramics: one obtained from glasses prepared by a standard melt-quenching technique and the other derived from amorphous powders obtained by means of the ball-milling process. In the study we compare and evaluate the impact of B_2_O_3_ additive on the glass-formation/amorphization and crystallization in the Li_2_O-Al_2_O_3_-B_2_O_3_-TiO_2_-P_2_O_5_ systems as a function of the preparation route. For this purpose, we employ two experimental methods for structural investigations: X-ray diffractometry (XRD) and magic-angle spinning nuclear magnetic resonance (MAS NMR) spectroscopy.

## 2. Experimental

Glasses from the LAT(B)P_MQ series (see Table 1) were prepared by melting: Li_2_CO_3_ (Merck), NH_4_H_2_PO_4_ (POCh), anatase TiO_2_ (Merck), Al_2_O_3_ (Merck) and H_3_BO_3_ (POCh) at 1400 °C for 30 min in alumina crucible. Next, the melt was quenched between two stainless-steel plates. The glassy plates obtained were of 1–2 mm thick and opaque in color.

Subsequently, the LAT(B)P glassy plates were annealed at 900 °C for 12 h to enable crystallization. The samples after heat treatment are labelled as: LATP_MQ_ceramic, LATBP01_MQ_ceramic and LATBP03_ MQ_ceramic.

Three amorphous materials of the same nominal compositions were prepared by the ball-milling technique and labelled: LATP_BM_glass, LATBP01_BM_glass and LATBP03_BM_glass.

Stoichiometric amounts of chemicals (the same as in the case of melt-quenched glasses) were ground, placed in an alumina crucible and heated at 350 °C for 3 h to release the volatile products, namely water, ammonia and nitrogen oxides. The presynthesized mixtures were placed in tungsten carbide milling jars and ball-milled for 72 h at 600 rpm in ethanol as an immersion medium. After that, the final powders were dried under vacuum for 24 h at 50 °C, pelletized and annealed at 900 °C for 12 h to enable crystallization and allow the material to sinter. The ball-milled samples after sintering were labelled as: LATP_BM_ceramic, LATBP01_BM_ceramic and LATBP03_BM_ceramic.

Phase composition of the as-prepared glassy materials and ceramics after crystallization were examined by means of X-ray diffractometry. Diffractograms for LAT(B)P were collected at room temperature in the 2*θ* range from 15° to 80° with 0.05° step and counting rate of 0.5 s per step with CuKα line by means of a Philips X’PertPro diffractometer.

^27^Al, ^7^Li, ^31^P and ^11^B MAS NMR spectra were acquired on a 400 MHz Bruker AVANCE spectrometer, equipped with 4 mm HXY triple-resonance probe used in double resonance mode and spinning at a MAS frequency of 12.5 kHz. The Larmor frequencies of ^27^Al, ^7^Li, ^31^P and ^11^B isotopes were equal to: 104.3, 155.4, 161.9 and 128.4 MHz, respectively at 9.4 T. One- dimensional (1D) ^27^Al, ^7^Li, ^31^P and ^11^B NMR spectra were recorded using single-pulse experiments with pulse length corresponding to a tilt angle of π/8, π/8, π/6 and π/6, respectively as well as recycle delays of 2, 2, 120 and 2 s respectively. The number of scans ranged from 100 to 800 depending on the sample. ^27^Al, ^7^Li, ^31^P and ^11^B chemical shift values were referenced to 1 mol/L A_l_Cl_3_, 1 mol/L LiCl, 85 wt% H_3_PO_4_ and 0.3 mol/L H_3_BO_3_ aqueous solutions. The NMR spectra were simulated using dmfit software [48].

## 3. Results and Discussion

### 3.1. X-ray Diffraction (XRD) (Melt-Quenched Samples)

Figure 1 shows diffractograms for the powdered samples of LATP_MQ_glass, LATBP01_MQ_glass and LATBP03_MQ_glass. A broad amorphous halo, typical for structures with the lack of long-range ordering, is present for all three samples at ca. 25°. However, in the case of LATP_MQ_glass two weak reflections at ca. 24° and 20.3° are detected. These peaks coincide well with the two most prominent maxima of L(A)TP reference pattern (compare with Figure 2) which suggests that some detectable amount of LATP precipitations exists in the as-cast glass probably because the melt was cooled at an insufficiently low cooling rate. The lack of these reflections for the two other materials can be considered as evidence that the addition of B_2_O_3_ enhances the glass-forming properties of Li_2_O-Al_2_O_3_-TiO_2_-P_2_O_5_ system.

XRD studies for LAT(B)P_MQ series after heat-treatment at 900 °C for 12 h are displayed in Figure 2. Due to the heat-treatment, LATP_MQ_glass transforms into a ceramic material. A thorough analysis of its phase composition with the use of ICDD crystallographic database patterns, reveals that apart from the predominant NASICON-type LiTi_2_(PO_4_)_3_ phase (∘), other crystalline phases also crystallize. In particular, one can observe the formation of rutile TiO_2_ (✶) and cubic AlPO_4_ (◆). Trace amounts of the lithiated L(A)TP (!) and LiTiPO_5_ (+) phases can also be detected.

LATBP01_MQ_ glass undergoes a crystallization process which results in the formation of the main, NASICON-type LiTi_2_(PO_4_)_3_. It is accompanied by minor secondary phases: cubic AlPO_4_, lithiated L(A)TP (!) and LiTiPO_5_ (+). When compared to boron-free LATP_MQ_ceramic, LATBP03_MQ_ceramic sample contains lower amounts of unwanted phases. Only orthorhombic AlPO_4_ (✶) can be detected.

In all of the presented XRD patterns, the predominant phase after crystallization is LATP. However, closer analysis of XRD results, especially the content of the other phases formed during annealing, indicate that for the boron incorporated LATP materials the concentration of foreign phases is lower than in the case of boron-free LATP ceramic.

### 3.2. XRD (Ball-Milled Samples)

Figure 3, Figure 4 and Figure 5 present three stages of LAT(B)P_BM series formation. It can be seen that after the heat treatment of reagents at 350 °C for 3 h, the phase composition for all samples is very similar (Figure 3a, Figure 4a and Figure 5a). The main diffraction peaks are ascribed to TiO_2_ (rutile) and Al_2_O_3_ (corundum). Because their melting point is in the range from 1800 to 2000 °C, they remain unaffected by the presynthesis conditions.

After ball-milling, however, it can be observed that the products are mainly amorphous with traces of remaining titania and newly formed LATP phase (Figure 3b, Figure 4b and Figure 5b). Moreover, by comparing diffractograms for all LAT(B)P_BM glasses series, one can notice that with the increased concentration of boron oxide, the amorphization proceeds more effectively, which is evidenced by broader and more pronounced amorphous halo and low-intensity diffraction peaks for LATBP01 (after BM) and LATBP03 (after BM).

Finally, when ball-milled samples are crystallized, the final glass-ceramics consist of the LATP phase with traces of secondary phases (mainly cubic AlPO_4_).

XRD investigations for LAT(B)P_BM glass-ceramics show that the boron addition affects the amorphization process considerably. The presence of B_2_O_3_ during ball-milling allows materials to be obtained with a lower amount of crystalline precipitation. On the other hand, B_2_O_3_ seems not to affect much the composition of final ceramics (after heat-treatment) with in all three cases consist mainly of the LATP phase with traces of AlPO_4_ (cubic).

### 3.3. Magic-Angle Spinning Nuclear Magnetic Resonance (MAS NMR) (Melt-Quenched Samples)

The ^27^Al MAS NMR spectrum of the LATP glass (Figure 6a) consists of wide asymmetric signal, typical of glassy samples. The asymmetric line-shape is due to the distribution of NMR parameters (chemical shit, quadrupolar constant, asymmetry parameter). Nevertheless, the spectrum enables the presence of three contributions to be detected at: 37.0, 6.6 and −15.6 ppm, corresponding to tetrahedral (Al^IV^), pentahedral (Al^V^) and octahedral (Al^VI^) coordination of aluminum, respectively [30,49,50,51,52].

For ceramic LATP material (Figure 6b) one can distinguish two separated lines. Both of them are narrower than in the case of the glassy material. Deconvolution of the spectra revealed the lines at: 38.6, 32.8, −15.3, −17.7 and −21.4 ppm. The signal at 38.6 ppm is assigned to AlPO_4_ compound, which was also detected by X–ray diffraction method [26,30,50,51]. The resonance at 32.8 ppm may be produced by a glassy phase containing Al^3+^ ions. In the region of AlO_6_ sites, two resonances at −15.3 and −17.7 ppm are characteristic for aluminum coordination in the NASICON–type phase [30], meaning that two LATP phases with similar chemical composition may coexist. The last signal at −21.4 ppm might be ascribed to aluminum in octahedral coordination in the LiAlP_2_O_7_ phase [53].

For the boron-incorporated LATBP glasses (Figure 6c,e), the same signals could be observed as for the LATP glass. However, if one considers the ceramic LATBP materials, more discrepancies could be noticed. For LATBP0.1 (Figure 6d) material, we can observe a significant reduction of the relative integrated intensity of the signal attributed to AlPO_4_ phase and the lack of residual glassy phase. In the case of LATBP0.3 (Figure 6f), no residual glassy phase is detected and the relative integrated intensity of the signal at −15.3 and −17.7 ppm, assigned to NASICON–type phase, is higher than in LATP ceramic, while the relative amount of the AlPO_4_ and LiAlP_2_O_7_ phases is nearly the same (see Table 2).

The obtained ^27^Al MAS NMR results are consistent with X–ray diffractograms. In the ceramic LATP and LATBP0.3 materials, excepted the LATP phase, a significant quantity of unwanted phases could be noticed. The LATBP0.1 material is characterized by the highest content of NASICON–type phase among all studied materials.

^31^P MAS NMR spectra for glassy LAT(B)P_MQ materials are dominated by broad, asymmetric signals, centered around −18.2 and −22.9 ppm for: LATP, LATBP0.1 and LATBP0.3 samples, respectively (Figure 7a,c,e). I. d’Anciães Almeida Silva et al. [54] also reported for LGGP glassy sample broad signal around −25 ppm.

For crystallized LAT(B)P_MQ samples, narrower resonances could be distinguished (Figure 7b,d,f). In the case of glass–ceramic LATP (Figure 7b), asymmetric resonance at −27.5 ppm could be deconvoluted into Gaussian/Lorentzian lines with isotropic chemical shifts of: −30.5, −27.9, −27.5, −26.5, −25.9, −25.0 and −24.1. The signal at −30.5 ppm can be assigned to phosphorus in crystalline AlPO_4_ phase [30,55]. The resonance at −22.6 could be due to the residual glassy phase. Subsequent overlapping lines may be attributed to two LATP phases with similar chemical composition. The resonances at −27.9 and −26.5 ppm correspond to phosphorus P(OTi)_4_ and P(OTi)_3_(OAl)_1_ environments in LATP (I) phase. While, the peaks at −27.5, −25.9, −25.0 and −24.1 ppm can be assigned, respectively to P(OTi)_4-n_(OAl)_n_ (n = 0, 1, 2 and 3) bands in LATP (II) phase [26,30,51,55]. It is worth mentioning that ^27^Al MAS NMR results also revealed two Al^3+^ environments, suggesting coexistence of two LATP phases. The ^31^P MAS NMR spectrum for LATP_ceramic_MQ exhibits also a signal at −22.6 ppm, ascribed to residual glassy phase. Moreover, except for the aforementioned signals, two additional resonances at −9.8 and −6.4 ppm could be discerned. They can be attributed to LiTiPO_5_ and Li_4_P_2_O_7_ phases, respectively [51,56,57].

For the boron-incorporated glass-ceramic materials, the signals assigned to LiTiPO_5_ and Li_4_P_2_O_7_ phases nearly vanished. Also, the intensity of the peak attributed to Al^3+^ in AlPO_4_ crystalline phase (or the residual glassy phase) decreased.

Additionally, for LATBP0.3 sample, one can notice a broad signal at −14.0 ppm, which corresponds to phosphorus environment in some unknown phase. In LATBP samples, the same P(OTi)_4−n_(OAl)_n_ (n = 0, 1, 2 and 3) structural units can be found. Their isotropic chemical shifts differ slightly.

More discrepancies can be observed when the relative integrated intensities of the components included in the LATP phase are compared. First of all, the signal assigned to P(OTi)_4_ environment is weaker for LATBP samples. Simultaneously, the relative integrated intensities of the peaks assigned to P(OTi)_3_(OA_l_)_1_, P(OTi)_2_(OA_l_)_2_ and P(OTi)_1_(OA_l_)_3_ units increased. Therefore, ^31^P MAS NMR results may suggest that more structural units containing Al^3+^ ions are formed during sintering of B–incorporated LATP materials as a result of partial substitution of Ti^4+^ by Al^3+^ in LATP phases. Comparing the relative integrated intensities of phosphorus P(OTi)_4_−n(OA_l_)n (n = 0, 1, 2 and 3) structural units in LATP phases, one can deduce the ratio of aluminum ions in respect to titanium ones, using the expression [26,30]:(1)$$\frac{Al^{3+}}{Ti^{4+}}=\frac{4I_{4}+3I_{3}+2I_{2}+I_{1}}{4I_{0}+3I_{1}+2I_{2}+I_{3}}=\frac{x}{2-x}$$

In the formula, *I_n_* (*n* = 0, 1, 2, 3 and 4) represents relative integrated signal intensities of 31P bands, which correspond to P(OTi)_4−n_(OA_l_)_n_ environments. Table 3 presents nominal and actual concentration of Al^3+^ in LATP phase. It can be noted that LATP sample without boron is characterized by low aluminum content, due to formation of some other aluminophosphate phases. Introduction of boron into the LATP material, led to higher concentration of aluminum ions in LATP phase, even above the nominal value of 0.3. This may be the result of the technological process during which the substrates were melted in alumina crucible. It is very likely that some Al^3+^ diffused from the crucible to the material during melting of the substrates at high temperatures.

^31^P MAS NMR results show that introduction of boron to the lithium aluminum titanium phosphate increased the concentration of P(OTi)_3_(OAl)_1_, P(OTi)_2_(OA_l_)_2_ and P(OTi)_1_(OA_l_)_3_ structural units leading to closer to nominal content of Al^3+^ in LATP phase and reduced the content of other phosphate phases. Those results are consistent with ^27^Al MAS NMR investigations, which showed that more aluminum is in the LATP phase when boron is present in the sample.

The central transition of ^7^Li MAS NMR spectra for glass and glass–ceramic materials are presented in Figure 8. The former are characterized by a single signal at −0.7 ppm, while the latter exhibit at least three resonances.

For ceramic LATP material two overlapping lines at −1.1 and −0.9 ppm could be assigned to Li1 and Li3 sites in NASICON-type structure, respectively [1,4,5,6,13]. The relative integrated intensity of the signal attributed to Li3 site is higher than Li1, which suggests that Li3 site is preferred for lithium ions. Additionally, in the ^7^Li NMR spectra two weaker signals at −0.5 and 0.2 ppm could be noticed. They may be ascribed to residual glassy phase.

For boron incorporated ceramics, resonance assigned residual glassy phase is no longer present. While the signal attributed to Li environment in LiAlP_2_O_7_ compound remains nearly the same. More changes occur in NASICON–type material. First of all, a slight shift of both peaks into more positive values can be observed. Also, in comparison to LATP material, the relative integrated intensities of both signals change considerably. For LATBP0.1 and LATBP0.3 ceramics, the Li3 site is even more preferable for lithium ions than the Li1 site.

Figure 9 presents ^11^B MAS NMR spectra for glassy LATBP materials, which consist of three signals at: 5.7, −2.5 and −3.7 ppm. Spectra for glassy materials are nearly identical. The peak at 5.7 ppm can be associated with BO_3_ site, while the remaining two can be assigned to the boron environment in BO_4_. The relative integrated intensity of the BO_4_ site is higher than for BO_3_, which suggests a higher concentration of BO_4_ sites in both glasses. 

In the case of the ceramic materials, one can observe changes in the boron environment depending on the concentration of boron in the material. For LATBP0.1 ceramic material, three signals corresponding to the BO_3_ site can be noticed at: 18.8, 7.7 and 3.3 ppm, while for BO_4_, only the peak at −1.4 ppm is present. The relative integrated intensity of the signals assigned to BO_3_ site is slightly higher than that of BO_4_.

The ^11^B MAS NMR spectra of the ceramic LATBP0.3 material also consist of 18.8 and 7.7 ppm signals. However, their intensity is much lower than in the case of LATBP0.1 material. Moreover, the peak at 3.3 ppm is no longer visible, while for BO_4_ sites, two resonances at: −1.2 and −3.3 ppm are found. Their high integrated intensity compared to intensity of the peaks assigned to BO_3_ sites suggests that boron in the BO_4_ environment is predominant for LATBP0.3 material. The assignment of these 11B resonances in ceramics samples is difficult owing to the lack of chemical shift reference for pure borophosphate materials. Considering the large number of sites, it can be suggested that boron probably substitutes partially aluminum in the ceramic materials.

### 3.4. Magic-Angle Spinning Nuclear Magnetic Resonance (MAS NMR) (Ball-Milled Samples)

The ^27^Al MAS NMR spectra for amorphous LATP and LATBP0.1 materials (Figure 10a,c) are composed of the wide signals centered around: 61, 36.7, 5.6, −17.2, −20.8 and −36 ppm. Despite the fact that the material should be amorphous, the resonances are much narrower than in glassy samples. However, full width at half maximum (FWHM) is large compared to ceramic materials, hence one can assume that amorphization of the materials is not complete, which was also suggested by XRD investigations. Signals at 61 and 36.7 ppm can be assigned to aluminum in tetrahedral coordination (AlO_4_), while that at 5.6 ppm to pentahedral coordination (AlO_5_) [30,49,50,51,52]. The remaining three resonances at: −17.2, −20.8 and −36 ppm are characteristic for octahedral coordination of aluminum (AlO_6_) [30,49,50,51,52].

Some significant changes are noticed on ^27^Al MAS NMR spectrum (Figure 10e) of the amorphous LATBP03_glass_BM sample. First of all, there are only five resonances at: 11.6, 5.2, −12.6, −16.8 and −23 ppm. Additionally, the relative integrated intensity of the peaks related to aluminum in the NASICON-type phase (−16.8 and −23 ppm) is lower than in the case of LATP_glass_BM or LATBP01_glass_BM materials. Therefore, introduction of a higher content of boron into LATP resulted in lower precipitation of NASICON¬–type phase in this stage of ball-milling synthesis.

The ^27^Al MAS NMR spectrum for ceramic LATP material (Figure 10b) is composed of narrow peaks at 40.5, −9.6, −14.3, −15.7 and −20.1 ppm. The signal at 40.5 ppm is characteristic for AlPO_4_ compound in AlO_4_ coordination of aluminum [30,49,50,51,52]. The resonances located at −14.3 and −15.7 ppm correspond to aluminum in AlO_6_ environment in two slightly different LATP phases [53,54,58]. The occurrence of two LATP phases was also observed by C. Vinod Chandran in Li_1+x_A_lx_Ti_2–x_(PO_4_)_3_ samples with x > 0.5 [58]. The peak at −20.1 ppm can be attributed to aluminum environment in LiAlP_2_O_7_ compound [53,57]. Furthermore, the weak signal observed at −9.6 ppm may be assigned to some other unknown phase containing Al ions (see Table 4).

For boron–incorporated LATBP ceramics (Figure 11d,f), the aforementioned signals are still observed. Moreover, one can notice additional weak signal at 37.4 ppm, which corresponds to some unknown phase. It is worth mentioning that relative integrated intensities of the signals assigned to LATP phases decrease and simultaneously cause the growth of AlPO_4_ content in materials with an increasing amount of boron. This may suggest that introduction of too much boron into the LATP material results in the formation of higher content of AlPO_4_ phase, which is detrimental for the mobility of lithium ions.

Figure 11a,c,f presents ^31^P MAS NMR spectra for amorphous LATP, LATBP01 and LATBP03 materials. Spectra for LATP and LATBP01 are nearly the same. They are characterized by an asymmetric peak at −27.3 ppm with significant widening of the signal for lower intensities. The deconvolution of spectra resulted in the occurrence of six overlapping Lorentzian line-shapes around: −37.5, −27.6, −26.2, −22.0, −15.0 and −7.0 ppm. Wide signals at −37.5, −15.0 and −7.0 ppm may be attributed to precursors of AlPO_4_, LiAlP_2_O_7_ and LiTiPO_5_ compounds, respectively; while the resonances at −27.6, −26.2 and −22.0 ppm can be, respectively, assigned to the phosphorus environment in P(OTi)_4−n_(OAl)_n_ (n = 0, 1 and 2) [26,30,51,56]. More significant changes may be observed for amorphous LATBP03 material. First of all, a wide asymmetric peak around −11 ppm may be found. Except for the lines included in this peak, one can also observe three further narrower signals at −30.0, −29.7 and −27.5 ppm. The signals at −30.0 and −29.7 ppm might be assigned to AlPO_4_ phase, while the one at −27.5 ppm to LATP compound. Therefore, when considering the wide signal and the low content of NASICON-type and aluminophosphate compounds, we can conclude that the amorphization of the LATBP03 sample is higher than in the case of LATP and LATBP01, which was also observed in XRD investigations.

For the ceramic LATP (Figure 11b) material obtained by sintering of the amorphous one, ^31^P MAS NMR spectrum consist of the signals located at: −30.2, −27.6, −27.2, −26.5, −26.0, −25.3, −24.4 and −22.9 ppm. The peaks at −30.2 and −22.6 ppm are characteristic for phosphorus in AlPO_4_ and LiAlP_2_O_7_ compounds, respectively [30,51,53,56,57]. The overlapping lines could be divided into two groups, which may be assigned to LATP_1 and LATP_2 phases. The resonances at −27.6 and −26.5 ppm correspond to LATP_1. The remaining signals around −27.2, −26.0, −25.3 and −24.4 ppm can be attributed to the phosphorus environment in P(OTi)_4−n_(OAl)_n_ (n = 0, 1, 2 and 3) in the LATP_2 phase. Their relative integrated intensity is higher than in the case of the LATP_1 phase [26,30,51,56].

For boron-incorporated ceramics (Figure 11d,f), the same signals may be observed. Additionally, the wide one at −14.0 ppm can be also noticed, which is assigned to some unknown phosphate phase. Despite this additional resonance, the shape of the main peak for all of the studied ceramics remains nearly the same.

However, further analysis of the ^31^P MAS NMR results, especially that concerning relative integrated intensities of the signal assigned to P(OTi)_4−n_(OAl)_n_ (n = 0, 1, 2 and 3) bands in NASICON-type phases may provide some additional information about the chemical composition of the LATP. The ratio of Al ions to Ti ones, can be deduced using the following formula [26,30]:(2)$$\frac{Al^{3+}}{Ti^{4+}}=\frac{4I_{4}+3I_{3}+2I_{2}+I_{1}}{4I_{0}+3I_{1}+2I_{2}+I_{3}}=\frac{x}{2-x}$$
where *I_n_* (*n* = 0, 1, 2, 3 and 4) stands for relative integrated signal intensities of phosphorus P(OTi)_4−n_(OAl)_n_ environments. Comparison of the actual and nominal values of aluminum content in LATP material is presented in Table 5. For LATP and LATBP01 ceramics, the concentration of Al^3+^ in NASICON–type phase is about 0.3, which agrees with the nominal value, while for the ceramic material with higher content of boron, the actual deducted value of aluminum ions inside LATP is insignificantly lower, equal to 0.27.

Summing up, ^31^P MAS NMR results show that too high a content of boron in the ceramic sample may not be attractive to the formation of the NASICON-type phase, due to the occurrence of some other phosphates and lower than nominal value of aluminum ions in LATP.

^7^Li MAS NMR spectra of the ball-milled amorphous materials are presented in Figure 12a,c,e. In the case of LATP and LATBP01 samples, the ^7^Li resonance could be simulated by one wide peak around −0.8 ppm; while for the LATBP03 material, the deconvolution of the CT can be deconvoluted into three signals at: −0.6, −0.2 and 0 ppm. The most intense peak is the one located at −0.6 ppm.

The spectra of the ceramic materials (Figure 12b,d,f) are also characterized by wide resonance. For the LATP sample, it can be deconvoluted into three lines located at −1.0, −0.9 and 0.1 ppm. The latter signal may be associated with lithium in LiAlP_2_O_7_ compound, while the former can be assigned to lithium sites (respectively Li1 and Li3) in NASICON structure [30,51,52,59]. The relative integrated intensity of the signal attributed to Li3 site is higher than for the resonance for Li1 site, which may suggest that Li^+^ prefer occupation of Li3 sites. A similar observation may be made for the LATBP03 sample. However, in the case of LATBP01 material, the ^7^Li resonance is much wider and resonances are slightly shifted toward positive values of ppm (−0.2, −0.1 and 0.4 ppm).

Figure 13 presents ^11^B MAS NMR spectra for amorphous and ceramic materials. For LATBP0.1 amorphous sample, only one resonance at −3.6 ppm could be observed. It was assigned to the boron environment in BO_4_, while for LATBP0.3 material, the deconvolution of the ^11^B MAS NMR spectrum resulted in four signals at: 18.5, −1.5, −3.5 and −4.1 ppm. The peak at 18.5 ppm can be assigned to BO_3_ group and the latter ones (−1.5, −3.5 and −4.1 ppm) may be attributed to BO_4_ group. 

In both cases, the relative integrated intensity of the signals associated with the BO_4_ environment is higher than for the BO_3_ one, suggesting a higher content of BO_4_ groups in amorphous materials. Considering the ceramic samples, no BO_3_ groups can be observed and the deconvolution of the ^11^B MAS NMR spectra revealed the occurrence of four overlapping lines at: −0.7, −3.6, −4.5 and −8.0 ppm.

## 4. Conclusions

In this paper two families of boron-incorporated LATP glasses were obtained by two different synthesis routes. After crystallization via a heat-treatment procedure, ceramic materials were obtained and their structure and phase content were then compared by XRD and MAS-NMR.

It can be concluded that:XRD studies show that the addition of B_2_O_3_ improves the glass-forming properties of LATP-based glasses prepared by the melt-quenching method (as evidenced by the lower amount of crystalline precipitation) and results in more effective amorphization of LAT(B)P materials during the ball-milling process (the amount of crystalline precipitation is considerably reduced due to boron addition).XRD studies show that the crystallization of LAT(B)P glasses leads to the formation of LATP phase, which is predominant in all studied materials. However, for the boron-incorporated materials, the concentration of foreign phases is lower than in the case of boron-free LATP ceramic.MAS NMR results are consistent with XRD studies and show that LATBP01_MQ material is characterized by the highest content of LATP phase among all studied materials.MAS NMR studies also show that the introduction of boron to the lithium aluminum titanium phosphate increases the concentration of P(OTi)_3_(OAl)_1_, P(OTi)_2_(OAl)_2_ and P(OTi)_1_(OAl)_3_ structural units leading to a content closer to nominal content of Al3+ in LATP phase and a reduced content of other phosphate phases.MAS NMR allows two LATP phases of slightly different chemical composition to be distinguished. This suggests that LATP grains may not be homogeneous, where regions closer to grain boundaries have different chemical composition than grain interiors.According to the MAS NMR results, boron in the BO_4_ environment is predominant for all MQ and BM materials.

The results obtained in this paper show that the amorphization and heat treatment of boron-incorporated LATP materials can be an effective way of obtaining LATP ceramics for all-solid state batteries. However, a remaining challenge is to carefully choose the heat-treatment conditions in order to reduce the precipitation of unwanted phases which can potentially hinder the Li^+^ transport. Further research is needed in order to optimize the precipitation process in order to obtain phase-pure LATP glass-ceramics for application in all-solid state batteries.

## Figures and Tables

**Figure 1 nanomaterials-11-00390-f001:**
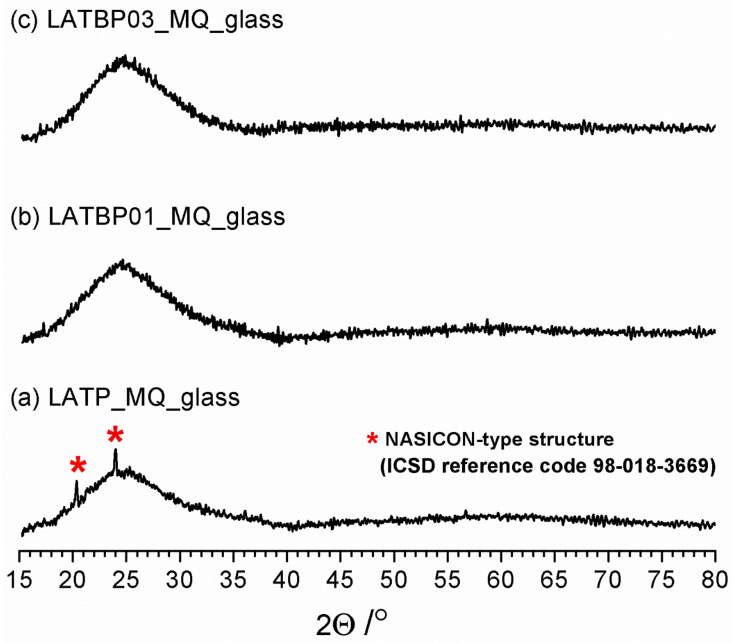
Diffractograms for LAT(B)P_MQ_glass series. Traces of Li_1.3_A_l0_._3_Ti_1.7_(PO_4_)_3_ (LATP) precipitations are marked by asterisks. (**a**)LATP_MQ_glass (**b**) LATBP01_MQ_glass (**c**) LATBP03_MQ_glass.

**Figure 2 nanomaterials-11-00390-f002:**
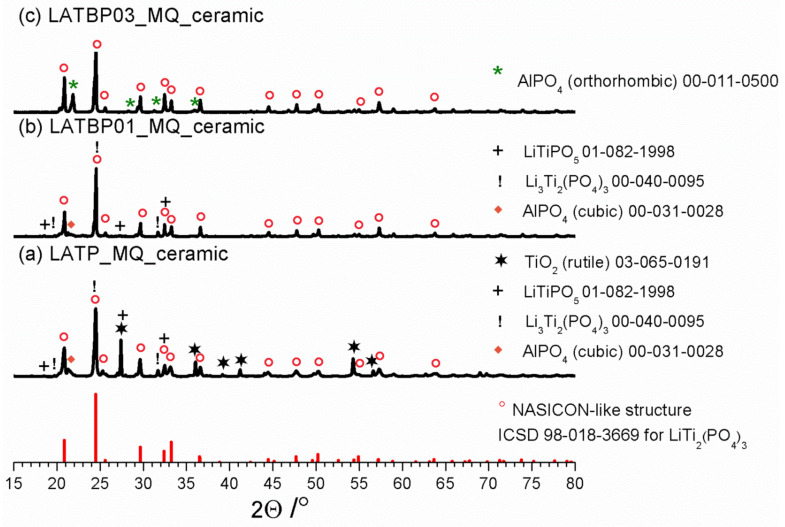
Diffractograms for LAT(B)P_MQ_ceramic series. For comparison ICDD pattern for NASICON-type structure is shown. Peaks ascribed to secondary phases are marked by symbols (see legend). (**a**) LATP_MQ_ceramic (**b**) LATBP01_MQ_ceramic (**c**) LATBP03_MQ_ceramic.

**Figure 3 nanomaterials-11-00390-f003:**
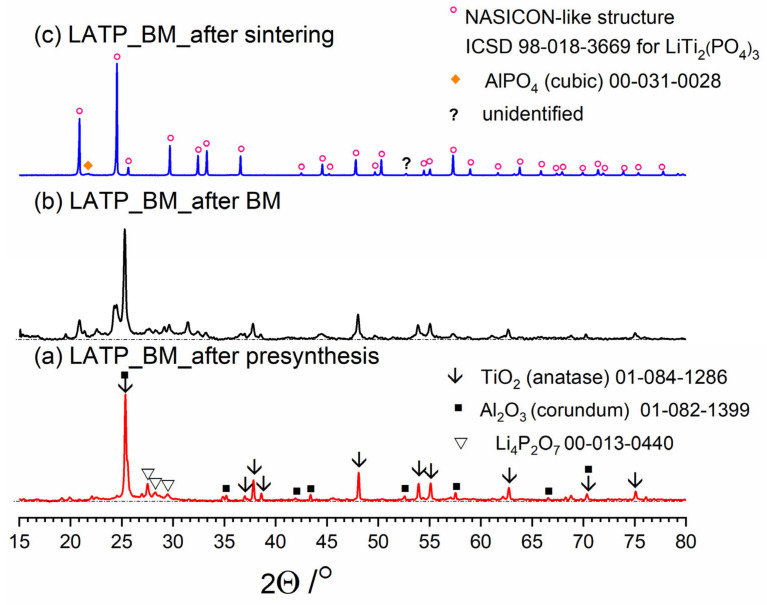
Diffractogrrams presenting different stages of LATP_BM_ceramic formation: (**a**) after presynthesis at 350 °C for 3 h; (**b**) after ball-milling at 600 rpm for 72 h; (**c**) after heat treatment at 900 °C for 12 h.

**Figure 4 nanomaterials-11-00390-f004:**
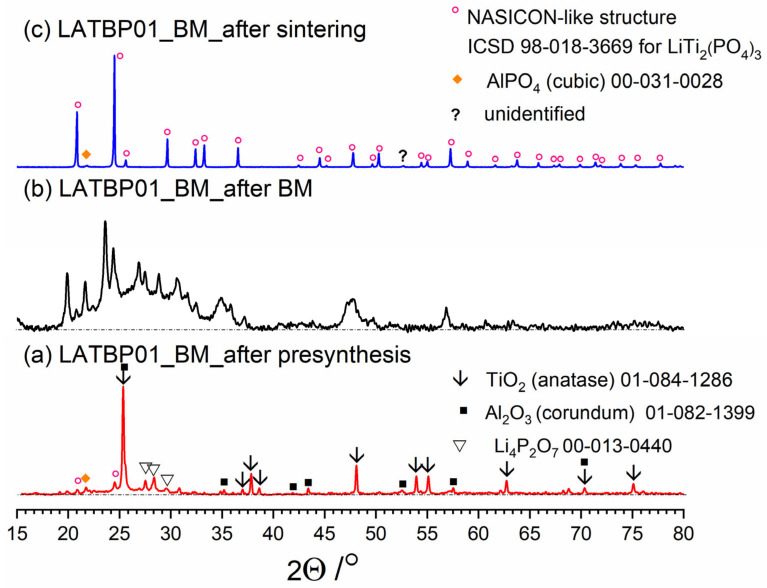
Diffractograms presenting different stages of LATBP01_BM_ceramic formation: (**a**) after presynthesis at 350 °C for 3 h; (**b**) after ball-milling at 600 rpm for 72 h; (**c**) after heat treatment at 900 °C for 12 h.

**Figure 5 nanomaterials-11-00390-f005:**
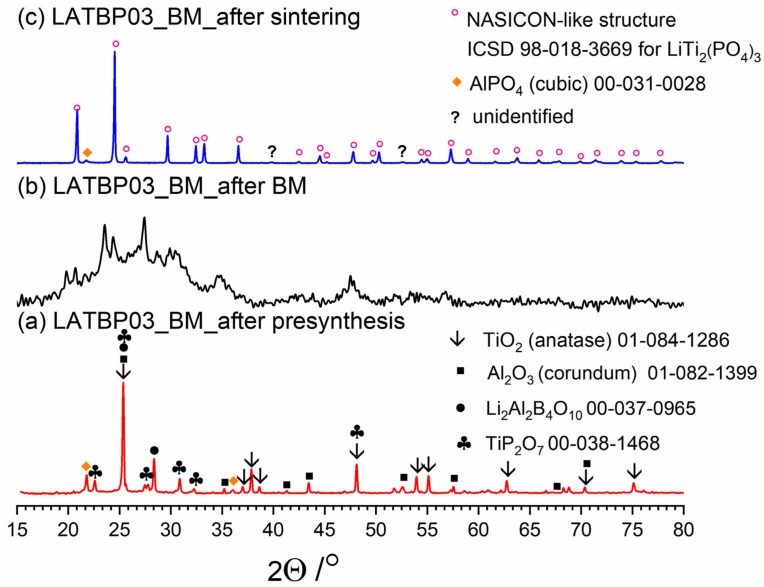
Diffractograms presenting different stages of LATBP03_BM_ceramic formation: (**a**) after presynthesis at 350 °C for 3 h; (**b**) after ball-milling at 600 rpm for 72 h; (**c**) after heat treatment at 900 °C for 12 h.

**Figure 6 nanomaterials-11-00390-f006:**
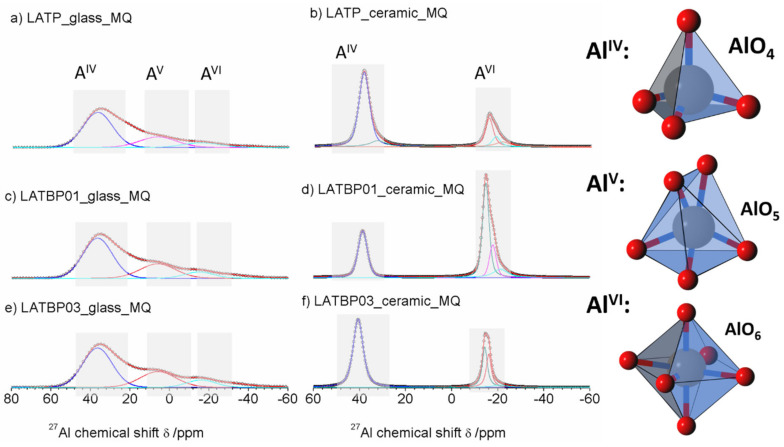
1D ^27^Al magic-angle spinning nuclear magnetic resonance (MAS NMR) spectra of the LAT(B)P_MQ series: (**a**) LATP_glass_MQ (**b**) LATP_ceramic_MQ (**c**) LATBP01_glass_MQ (**d**) LATBP01_ceramic_MQ (**e**) LATBP03_glass-MQ (**f**) LATBP03_ceramic_MQ Peaks corresponding to the environments: AlO_4_, AlO_5_ and AlO_6_ are denoted by: Al^IV^, Al^V^ and Al^VI^, respectively.

**Figure 7 nanomaterials-11-00390-f007:**
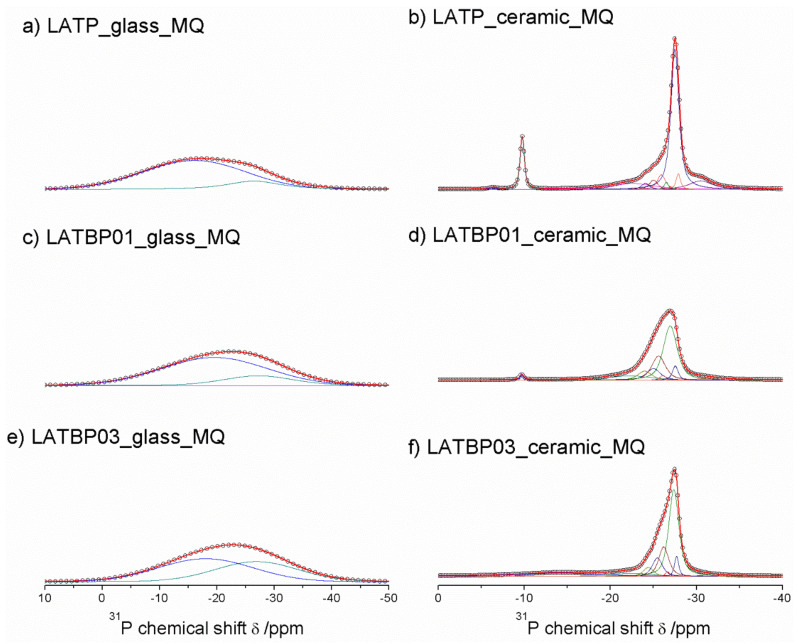
**1D**^31^P MAS NMR spectra of the LAT(B)P_MQ series. (**a**) LATP_glass_MQ (**b**) LATP_ceramic_MQ (**c**) LATBP01_glass_MQ (**d**) LATBP01_ceramic_MQ (**e**) LATBP03_glass-MQ (**f**) LATBP03_ceramic_MQ.

**Figure 8 nanomaterials-11-00390-f008:**
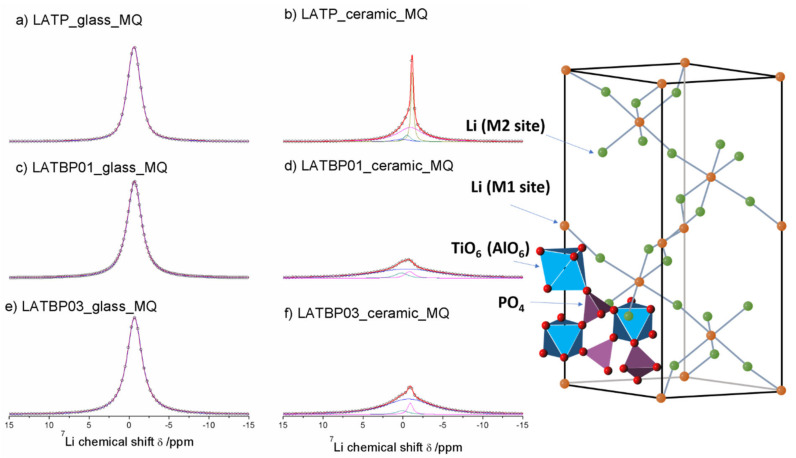
1D ^7^Li MAS NMR spectra of the LAT(B)P_MQ series. (**a**) LATP_glass_MQ (**b**) LATP_ceramic_MQ (**c**) LATBP01_glass_MQ (**d**) LATBP01_ceramic_MQ (**e**) LATBP03_glass-MQ (**f**) LATBP03_ceramic_MQ.

**Figure 9 nanomaterials-11-00390-f009:**
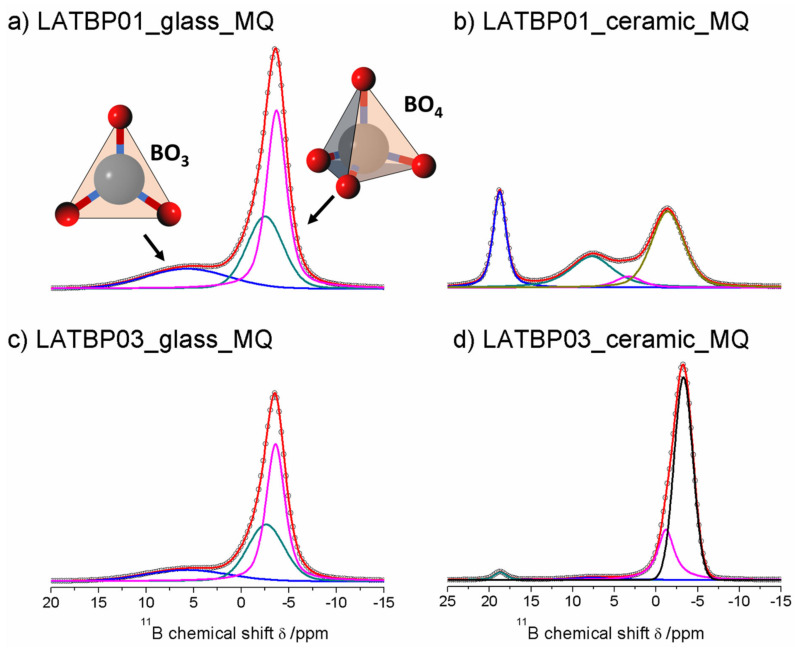
1D ^11^B MAS NMR spectra of the LAT(B)P_MQ series. (**a**) LATBP01_glass_MQ (**b**) LATBP01_ceramic_MQ (**c**) LATBP03_glass_MQ (**d**) LATBP03_ceramic_MQ.

**Figure 10 nanomaterials-11-00390-f010:**
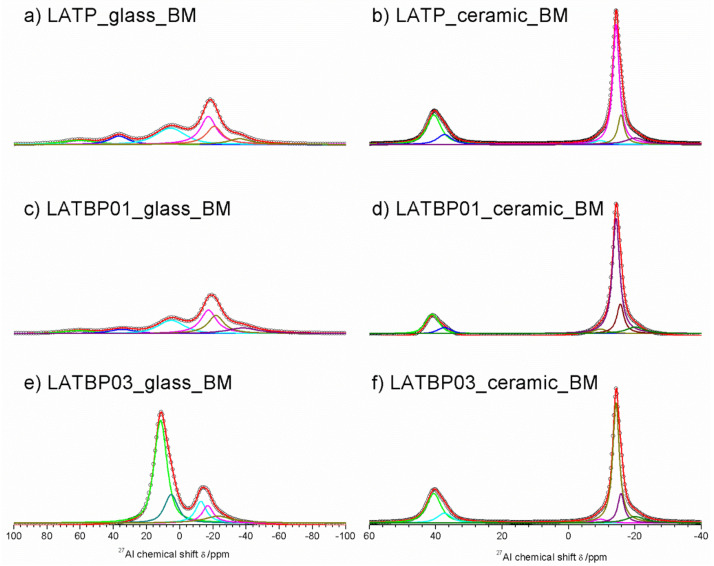
1D^27^Al MAS NMR spectra of the LAT(B)P_BM series. (**a**) LATP_glass_BM (**b**) LATP_ceramic_BM (**c**) LATBP01_glass_BM (**d**) LATBP01_ceramic_BM (**e**) LATBP03_glass_BM (**f**) LATBP03_ceramic_BM.

**Figure 11 nanomaterials-11-00390-f011:**
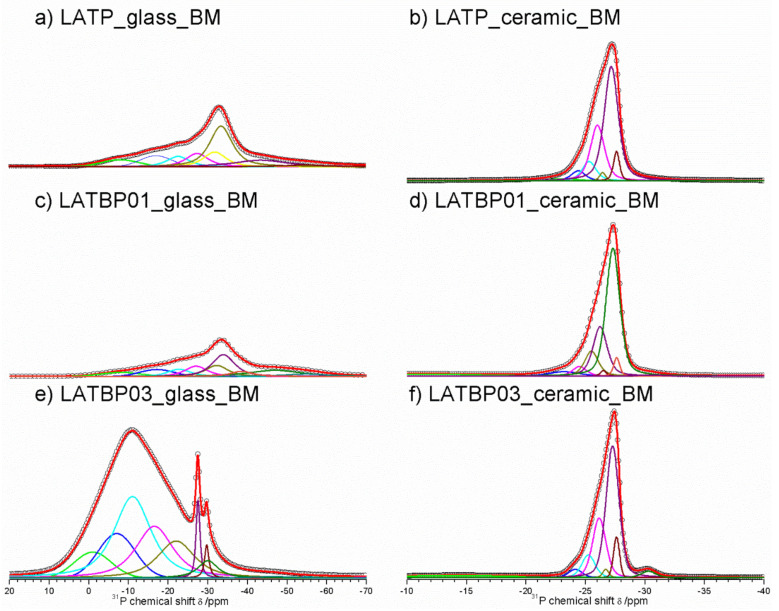
**1D**^31^P MAS NMR spectra of the LAT(B)P_BM series. (**a**) LATP_glass_BM (**b**) LATP_ceramic_BM (**c**) LATBP01_glass_BM (**d**) LATBP01_ceramic_BM (**e**) LATBP03_glass_BM (**f**) LATBP03_ceramic_BM.

**Figure 12 nanomaterials-11-00390-f012:**
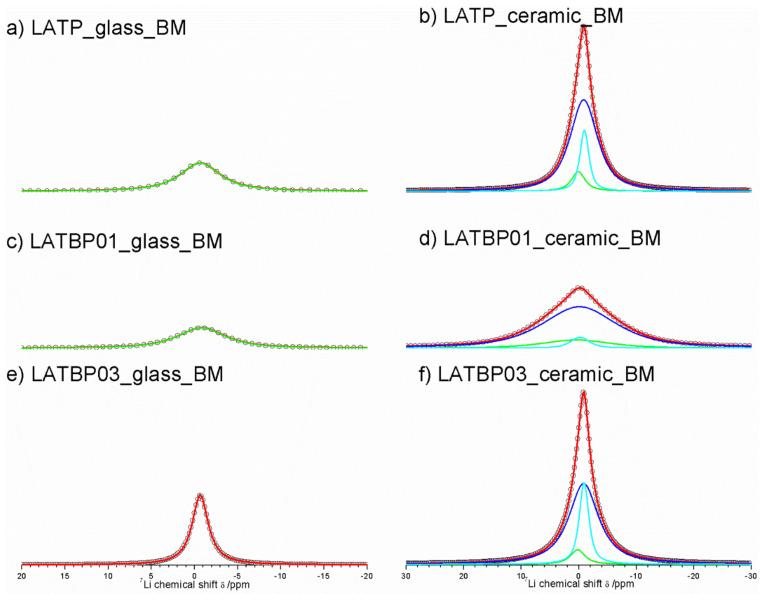
1D ^7^Li MAS NMR spectra of the LAT(B)P_BM series. (**a**) LATP_glass_BM (**b**) LATP_ceramic_BM (**c**) LATBP01_glass_BM (**d**) LATBP01_ceramic_BM (**e**) LATBP03_glass_BM (**f**) LATBP03_ceramic_BM.

**Figure 13 nanomaterials-11-00390-f013:**
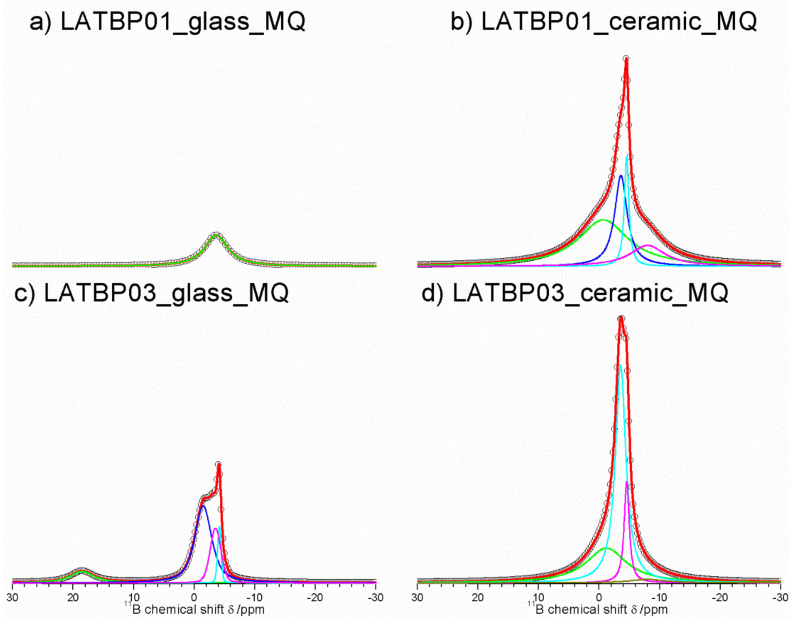
**1**D ^11^B MAS NMR spectra of the LAT(B)P_BM series. (**a**) LATBP01_glass_BM (**b**) LATBP01_ceramic_BM (**c**) LATBP03_glass_BM (**d**) LATBP03_ceramic_BM.

**Table 1 nanomaterials-11-00390-t001:** Nominal compositions of the LAT(B)P_MQ and _BM glass series.

GLASS CODE	Li_2_O	Al_2_O_3_	B2O_3_	TiO_2_	P2O_5_
LATP_GLASS	16.25	3.75	0	42.5	37.5
LATBP01_GLASS	17.5	3.75	1.25	40	37.5
LATBP03 _GLASS	20	3.75	3.75	35	37.5

**Table 2 nanomaterials-11-00390-t002:** Relative integrated intensities (II), full widths at maximum height (FWHM) and isotropic chemical shifts (δ) of the peaks used to simulate glass-ceramic ^27^Al MAS NMR shown in Figure 6.

	AlO_4_ (AlPO_4_)			AlO_5_ (Residual Glassy Phase)			AlO_6_ (LATP_1)			AlO_6_ (LATP_2)			AlO_6_ (LiAlP_2_O_7_)		
Sample Code	δ [ppm]	FWHM	II [%]	δ [ppm]	FWHM	II [%]	δ [ppm]	FWHM	II [%]	δ [ppm]	FWHM	II [%]	δ [ppm]	FWHM	II [%]
LATP_MQ_CERAMIC	38.6	38.6	58.3	32.8	8	8.3	−15.3	4.1	16.4	−17.7	4.1	10.7	−21.4	7.7	5.4
LATBP0.1_MQ_CERAMIC	38.6	38.6	29.2				−14.9	3.5	48.7	−17.8	3.5	14.1	−21.3	7.7	8.0
LATBP0.3_MQ_CERAMIC	40.6	40.6	59.4				−14.6	2.8	21.0	−16.3	2.8	17.6	−21.3	7.7	2.0

**Table 3 nanomaterials-11-00390-t003:** Relative integrated intensities (II), full widths at maximum height (FWHM) and isotropic chemical shifts (δ) of the line-shapes used to simulate ^31^P MAS NMR spectra shown in Figure 7.

	**Composition**		**P(OTi)_4_ (I)**			**P(OTi)_4_ (II)**				**P(OTi)_3_(OAl)_1_ (I)**			**P(OTi)_3_(OAl)_1_ (II)**		
**Sample Code**	**x_NOM_**	**x_NMR_**	**δ** **[ppm]**	**FWHM**	**II** **[%]**	**δ** **[ppm]**	**FWHM**		**II** **[%]**	**δ** **[ppm]**	**FWHM**	**II** **[%]**	**δ** **[ppm]**	**FWHM**	**II** **[%]**
LATP_MQ_CERAMIC	0.3	0.15	−27.9	0.6	3.2	−27.5	1.1		52.5	−26.5	0.6	1.4	−25.9	1.1	5.6
LATBP01_MQ_CERAMIC	0.3	0.36	−27.6	0.8	4.8	−27.0	1.9		43.5	−26.1	0.8	1.6	−25.6	1.9	19.3
LATBP03_MQ_CERAMIC	0.3	0.31	−27.7	0.6	4.4	−27.4	1.4		44.3	−26.6	0.6	1	−26.2	1.4	15.1
	**Composition**				**P(OTi)_2_(OAl)_2_ (II)**						**P(OTi)_1_(OAl)_3_ (II)**				
**Sample Code**	**x_NOM_**		**x_NMR_**		**δ**		**FWHM**		**II** **[%]**		**δ** **[ppm]**	**FWHM**		**II** **[%]**	
LATP_MQ_CERAMIC	0.3		0.15		−25.0		1.1		3.6		−24.1	1.1		2.3	
LATBP01_MQ_CERAMIC	0.3		0.36		−25.0		1.9		9.3		−24.0	1.9		7.3	
LATBP03_MQ_CERAMIC	0.3		0.31		−25.0		1.4		9.5		−24.4	1.4		4.7	
	**Composition**		**AlPO_4_**					**Residual Glassy Phase**				**LiTiPO_5_**			
**Sample Code**	**x_NOM_**	**x_NMR_**	**δ** **[ppm]**		**FWHM**		**II** **[%]**	**δ** **[ppm]**	**FWHM**		**II** **[%]**	**δ** **[ppm]**	**FWHM**	**II** **[%]**	
LATP_MQ_CERAMIC	0.3	0.15	−30.5		3.8		9.3	−22.6	5.8		11.3	−9.8	0.6	10.4	
LATBP01_MQ_CERAMIC	0.3	0.36	−31.0		3.6		2.2	−22.3	5.8		10.5	−9.8	0.6	1.5	
LATBP03_MQ_CERAMIC	0.3	0.31	−30.5		3.6		1.2	−22.6	5.8		6.3	-	-	-	

**Table 4 nanomaterials-11-00390-t004:** Relative integrated intensities (II), full widths at maximum height (FWHM) and isotropic chemical shifts (δ) of the line-shapes used to simulate ^27^Al MAS NMR spectra shown in Figure 10.

	AlO_4_ (AlPO_4_)			AlO_5_ or AlO_6_			AlO_6_ (LATP) (II)			AlO_6_ (LATP) (I)			AlO_6_ (LiAlP_2_O_7_)			AlO_4_		
Sample	δ [ppm]	FWHM	II [%]	δ [ppm]	FWHM	II [%]	δ [ppm]	FWHM	II [%]	δ [ppm]	FWHM	II [%]	δ [ppm]	FWHM	II [%]	δ [ppm]	FWHM	II [%]
LATP_MQ_CERAMIC	40.2	4.5	11.9	−9.6	4.8	3.5	−14.3	2.4	61.5	−15.8	2.4	16.3	−20.1	6.7	6.8			
LATBP01_MQ_CERAMIC	41.1	4.8	14.2	−9.6	4.8	4	−14.3	2.6	55.4	−15.6	2.6	14.2	−20.1	6.7	7.8	37.4	4.8	4.5
LATBP03_MQ_CERAMIC	40.6	5.2	23.7	−9.6	4.8	3	−14.3	2.4	46.9	−15.8	2.4	11.5	−20.1	6.7	7.1	37.4	5.2	7.9

**Table 5 nanomaterials-11-00390-t005:** Relative integrated intensities (II), full widths at maximum height (FWHM) and isotropic chemical shifts (δ) of the line-shapes used to simulate ^31^P MAS NMR spectra shown in Figure 11.

	Composition		P(OTi)_4_ (I)			P(OTi)_4_ (II)			P(OTi)_3_(OAl)_1_ (I)			P(OTi)_3_(OAl)_1_ (II)		
Sample Code	x_NOM_	x_NMR_	δ [ppm]	FWHM	II [%]	δ [ppm]	FWHM	II [%]	δ [ppm]	FWHM	II [%]	δ [ppm]	FWHM	II [%]
LATP_MQ_CERAMIC	0.3	0.30	−27.6	0.6	5.7	−27.2	1.4	51.4	−26.5	0.6	1.6	−26	1.4	24.9
LATBP01_MQ_CERAMIC	0.3	0.31	−27.6	0.6	3.2	−27.3	1.4	50	−26.5	0.6	1	−26.2	1.4	20.5
LATBP03_MQ_CERAMIC	0.3	0.27	−27.6	0.6	6	−27.3	1.3	47.5	−26.7	0.6	1.2	−26.2	1.3	21.8
	**Composition**				**P(OTi)_2_(OAl)_2_ (II)**					**P(OTi)_1_(OAl)_3_ (II)**				
**Sample Code**	**x_NOM_**		**x_NMR_**		**δ** **[ppm]**		**FWHM**	**II [%]**		**δ [ppm]**	**FWHM**		**II [%]**	
LATP_MQ_CERAMIC	0.3		0.30		−25.3		1.4	9.1		−24.4	1.4		4.9	
LATBP01_MQ_CERAMIC	0.3		0.31		−25.5		1.4	10.5		−24.4	1.4		4.3	
LATBP03_MQ_CERAMIC	0.3		0.27		−25.2		1.3	8.3		−24.2	1.3		2.7	
	**Composition**				**AlPO_4_**					**LiAlP_2_O_7_**				
**Sample Code**	**x_NOM_**		**x_NMR_**		**δ [ppm]**		**FWHM**	**II [%]**		**δ [ppm]**	**FWHM**		**II [%]**	
LATP_MQ_CERAMIC	0.3		0.30		−30.2		2.2	0.2		−22.9	4		2.3	
LATBP01_MQ_CERAMIC	0.3		0.31		−30.4		2.5	0.9		−22.9	4		3.8	
LATBP03_MQ_CERAMIC	0.3		0.27		−30.3		2	3.5		−22.8	4		2	

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
