# Peer review of "B2O3-Doped LATP Glass-Ceramics Studied by X-ray Diffractometry and MAS NMR Spectroscopy Methods"

_nanomaterials, 2021, doi:10.3390/nano11020390_

Round 1

Reviewer 1 Report

The reviewed paper titled “B2O3-doped LATP glass-ceramics studied by X-ray diffractometry and MAS NMR spectroscopy methods” presents scientifically sound data concerning the process of synthesis of an important group of ion-conducting oxides. The presented research focuses solely on XRD and NMR studies but provides important data for further steps of investigation of these materials. However, I believe this study can be better after some minor edits.

  • I find the abstract rather lengthy, I believe it can be shortened quite a bit without losing the main merit of the paper.
  • Figure 1 is redundant, does not provide any additional information, therefore, I advise it to be removed. The information about the color without the analysis of proper data does not provide information. This information can be moved to the discussion of the phases being formed during processing.
  • The NMR studies in the paper are analyzed in quite an extensive way. I believe the same should apply to XRD studies. The refinements of the diffractogram would provide additional information about for instance phase content and unit cell parameters. The information about the influence of the processing conditions on the structure of the material, investigated by XRD, would enhance the quality of the paper.
  • The XRD graphs of ball-milled samples 4b 5b 6b are really hard to interpret by a reader because of the low intensity of the diffraction peaks. How the XRD patterns where normalized? It’s hard to judge without the intensity scale on the graphs.

Author Response

Dear reviewer,

We would like to acknowledge the time and effort that you have put into assessing our manuscript entitled “B2O3-doped LATP glass-ceramics studied by X-ray diffractometry and MAS NMR spectroscopy methods” by Wioleta Ślubowska, Lionel Montagne, Olivier Lafon, François Méar, Konrad Kwatek.

In this revised version, we've given careful consideration to each comment in order to modify the manuscript in response to the points you have raised. A detailed item-by-item response to each of your points follows. For your convenience the modified parts of the manuscript are marked in blue.

With best regards,

Wioleta Ślubowska,

Reviewer #1:

The reviewed paper titled “B2O3-doped LATP glass-ceramics studied by X-ray diffractometry and MAS NMR spectroscopy methods” presents scientifically sound data concerning the process of synthesis of an important group of ion-conducting oxides. The presented research focuses solely on XRD and NMR studies but provides important data for further steps of investigation of these materials. However, I believe this study can be better after some minor edits.

Comment 1. I find the abstract rather lengthy, I believe it can be shortened quite a bit without losing the main merit of the paper.

Answer 1. We greatly appreciate that comment. To make the abstract more precise and comprehensive, we have shortened it.

Comment 2. Figure 1 is redundant, does not provide any additional information, therefore, I advise it to be removed. The information about the color without the analysis of proper data does not provide information. This information can be moved to the discussion of the phases being formed during processing.

Answer 2. Thank you for the comment. We decided to delete this part from the experimental section.

Comment 3. The NMR studies in the paper are analyzed in quite an extensive way. I believe the same should apply to XRD studies. The refinements of the diffractogram would provide additional information about for instance phase content and unit cell parameters. The information about the influence of the processing conditions on the structure of the material, investigated by XRD, would enhance the quality of the paper.

Answer 3. We agree that the Rietveld refinement would provide much more interesting information about the structure of the materials under study. However, to successfully refine the model it is required to have a pattern with good resolution and low background. In our case, we do not meet those criteria. More additional investigations, as well as preparation of new materials, should be performed to obtain excellent XRD patterns, which would greatly delay the publication of the paper.

Comment 4. The XRD graphs of ball-milled samples 4b 5b 6b are really hard to interpret by a reader because of the low intensity of the diffraction peaks. How the XRD patterns where normalized? It’s hard to judge without the intensity scale on the graphs.

Answer 4. Fig. 4, 5 and 6 represents the transformation of XRD patterns after specific technological processes. For the samples obtained just after ball-milling, we show that the products are mainly amorphous with some traces of TiO2 and LATP phases. Also, we show that the XRD peaks present for LATP material, begin to disappear with boron incorporation. This may suggest better amorphization of the material. The XRD patterns are now normalized and the broad peak could be clearly visible, especially for boron-LATP samples.

Reviewer 2 Report

This is an interesting study shedding light on the relevant preparation conditions that influence the morphology and properties of LATP-based solid electrolytes. Such materials are urgently needed for the development of new Li-based battery systems. 

The NMR analysis of the paper is fine although a bit to detailed in some aspects; the paper is relatively long and the authors should try to shorten it a bit to point out the main results a better way.

It would be great if the authors could provide some data on e.g. conductivity properties of the samples prepared and studied. Of course, a collaboration partner would be needed for that purpose.

Abstract: Please state directly in the abstract which NMR nuclei you used to study the samples, 11B, ... etc.

Author Response

Dear reviewer,

We would like to acknowledge the time and effort that you have put into assessing our manuscript entitled “B2O3-doped LATP glass-ceramics studied by X-ray diffractometry and MAS NMR spectroscopy methods” by Wioleta Ślubowska, Lionel Montagne, Olivier Lafon, François Méar, Konrad Kwatek.

In this revised version, we've given careful consideration to each comment in order to modify the manuscript in response to the points you have raised. A detailed item-by-item response to each of your points follows. For your convenience the modified parts of the manuscript are marked in blue.

With best regards,

Wioleta Ślubowska,

Reviewer #2:

This is an interesting study shedding light on the relevant preparation conditions that influence the morphology and properties of LATP-based solid electrolytes. Such materials are urgently needed for the development of new Li-based battery systems. 

The NMR analysis of the paper is fine although a bit to detailed in some aspects; the paper is relatively long and the authors should try to shorten it a bit to point out the main results a better way.

It would be great if the authors could provide some data on e.g. conductivity properties of the samples prepared and studied. Of course, a collaboration partner would be needed for that purpose.

Abstract: Please state directly in the abstract which NMR nuclei you used to study the samples, 11B, ... etc.

Answer: We greatly appreciate the comments. We had applied great importance on NMR analysis of the data to make sure that readers will fully understand the changes occurring during formation and sintering of the LAT(B)P materials.

Also, we agree that IS method could improve the presented results in the manuscript. However, we decided that this paper will be fully devoted to the structural investigations. The IS experiments of the materials under study will be performed later.

We included the NMR nuclei used to study the materials in the abstract.

Reviewer 3 Report

Recently, all-solid-state lithium-ion batteries (ASSBs) are in the spotlight as improved safety and increased energy density. It is important to choose appropriate solid electrolyte to ensure the good electrical performance and durability of ASSBs. The material with great interest is lithium aluminum titanium phosphate (LATP) because it has high thermal stability, non-flammability, great mechanical property, and high ionic conductivity. But LATP glasses have various phases with different conductivity. To solve this problem, this manuscript entitled “B2O3-doped LATP glass-ceramics studied by X-ray diffractometry and MAS NMR spectroscopy methods” describes that the comparable study of two families of LATP based glass-ceramics with different synthetic strategy: i) One is using a standard melt-quenching technique. ii) The other is amorphous powders with ball milling process. To evaluate the impact of the B2O3 additive on glass formation and crystallization, they did structural analysis using XRD and NMR method. I would like to recommend it to be published after a major revision.

Comment 1

The limitations of liquid electrolytes have been mentioned a lot in other papers, but I think it will help readers understand better if the author explains the advantages and uniqueness of solid electrolytes compared to liquid electrolytes in the introduction.

Comment 2

In the Experimental, the nominal composition of the samples was classified into three categories (LATP/ LATBP01/LATBP03), it should be clarified whether there is a reason for selecting these compositions. If the selection were made based on several candidates, it would be good to mention the reason why the other members were not selected.

Comment 3

The formation of amorphous structure to reduce boundary effect and unwanted products are interesting, however, the distinctive broad peak of amorphous structure appeared to be not as clear according to XRD data shown in figure 4,5, and 6.

Comment 4

The author has mentioned, “The presence of B2O3 during ball-milling allows to obtain materials with lower amount of crystalline precipitation”, by analyzing XRD data. The peak intensity shows the extent of crystallinity of the particular plane, as crystallinity is a relative term. In comparison with the available samples, total sum of the various prominent peaks intensity one can calculate the crystallinity. The calculation, such as Rietveld refinement, can be done using software.

Comment 5

In the results and discussion, the author noted the peak observed at 20.3 degrees (LATP_MQ) in Figure 2 as “probably because the melt was cooled at an insufficiently low cooling rate”. Is it possible to do not observe the peak when the cooling rate is increased? If there is relevant data, it would be nice to attach it.

Comment 6

Based on this paper, LATP with boron trioxide can inhibit the growth of unnecessary phases near the grain boundary, which has no conductivity. With the results of MAS NMR analysis, it could be guessed that the LATBP01 is adaptable as solid electrolyte, as it has the highest ion-conductivity respectively. To endue reliability for this result, further electrochemical test is recommended, such as EIS analysis.

Author Response

Dear reviewer,

We would like to acknowledge the time and effort that you have put into assessing our manuscript entitled “B2O3-doped LATP glass-ceramics studied by X-ray diffractometry and MAS NMR spectroscopy methods” by Wioleta Ślubowska, Lionel Montagne, Olivier Lafon, François Méar, Konrad Kwatek.

In this revised version, we've given careful consideration to each comment in order to modify the manuscript in response to the points you have raised. A detailed item-by-item response to each of your points follows. For your convenience the modified parts of the manuscript are marked in blue.

With best regards,

Wioleta Ślubowska,

Reviewer #3:

Recently, all-solid-state lithium-ion batteries (ASSBs) are in the spotlight as improved safety and increased energy density. It is important to choose appropriate solid electrolyte to ensure the good electrical performance and durability of ASSBs. The material with great interest is lithium aluminum titanium phosphate (LATP) because it has high thermal stability, non-flammability, great mechanical property, and high ionic conductivity. But LATP glasses have various phases with different conductivity. To solve this problem, this manuscript entitled “B2O3-doped LATP glass-ceramics studied by X-ray diffractometry and MAS NMR spectroscopy methods” describes that the comparable study of two families of LATP based glass-ceramics with different synthetic strategy: i) One is using a standard melt-quenching technique. ii) The other is amorphous powders with ball milling process. To evaluate the impact of the B2O3 additive on glass formation and crystallization, they did structural analysis using XRD and NMR method. I would like to recommend it to be published after a major revision.

Comment 1. The limitations of liquid electrolytes have been mentioned a lot in other papers, but I think it will help readers understand better if the author explains the advantages and uniqueness of solid electrolytes compared to liquid electrolytes in the introduction.

Answer 1. In the introduction part, we extended the text and included more advantages of the solid electrolytes.

Comment 2. In the Experimental, the nominal composition of the samples was classified into three categories (LATP/ LATBP01/LATBP03), it should be clarified whether there is a reason for selecting these compositions. If the selection were made based on several candidates, it would be good to mention the reason why the other members were not selected.

Answer 2. The LATP material is one of the promising candidates for application as solid electrolyte. Its preparation by a melt-quenching route requires very high temperatures. So then, we decided to modify the properties of the Li2O-Al2O3-TiO2-P2O5 glass with B2O3 additive. Boron trioxide, B2O3 has been used not only as an additive improving the glass-forming properties of the melt but also as an agent influencing the crystallization kinetics. We decided to add low amount of B2O3 additive (LATBP01) to investigate the structural changes. But also, we were curious, whether much higher amount of boron present in LATP material (LATBP03) will cause formation of another secondary phases.

Comment 3. The formation of amorphous structure to reduce boundary effect and unwanted products are interesting, however, the distinctive broad peak of amorphous structure appeared to be not as clear according to XRD data shown in figure 4,5, and 6.

Answer 3. The XRD patterns are now normalized and the broad peak could be clearly visible, especially for boron-LATP samples.

Comment 4. The author has mentioned, “The presence of B2O3 during ball-milling allows to obtain materials with lower amount of crystalline precipitation”, by analyzing XRD data. The peak intensity shows the extent of crystallinity of the particular plane, as crystallinity is a relative term. In comparison with the available samples, total sum of the various prominent peaks intensity one can calculate the crystallinity. The calculation, such as Rietveld refinement, can be done using software.

Answer 4. Thank you for your valuable comment that we totally agree. To perform a proper Rietveld refinement of the model it is required to have a pattern with good resolution and low background. In our case, we do not meet those criteria. More additional investigations, as well as preparation of new materials, should be performed to obtain excellent XRD patterns, which would greatly delay the publication of the paper.

Comment 5. In the results and discussion, the author noted the peak observed at 20.3 degrees (LATP_MQ) in Figure 2 as “probably because the melt was cooled at an insufficiently low cooling rate”. Is it possible to do not observe the peak when the cooling rate is increased? If there is relevant data, it would be nice to attach it.

Answer 5. We tried to maintain the same cooling rate for each sample using melt-quenching technique. However, even very small delay in cooling the melt may be crucial for precipitation of some phases. It is very plausible that for LATP sample, the temperature of the melt was not high enough. It could be observed that no additional peaks are present in the XRD patterns for LATP materials with B2O3 additive. Therefore, increasing the temperature required to melt the substrates may enough for obtaining the material without precipitations. We cannot prepare such experiment, because the temperature limit for our furnace is 1400°C.

Comment 6. Based on this paper, LATP with boron trioxide can inhibit the growth of unnecessary phases near the grain boundary, which has no conductivity. With the results of MAS NMR analysis, it could be guessed that the LATBP01 is adaptable as solid electrolyte, as it has the highest ion-conductivity respectively. To endue reliability for this result, further electrochemical test is recommended, such as EIS analysis.

Answer 6. Depending on the material, grain boundary may be composed of pores or other phases formed from the decomposition of the base material. For the former one, we agree that grain boundary has no conductivity and it is crucial to obtain the materials with high value of density. However, for the latter one, if some other secondary phases are formed, they may transport lithium ions. The value of σgb depends on the formed compounds, but it is very plausible that it will be lower than the conductivity of the base material. Taking into account the materials under study in the paper, indeed, the LATBP01 sample seems to be the most perspective one, due to the low amount of secondary non-conductive phases. But, one should keep in mind that the composition of the bulk also have impact on the value of conductivity. Therefore, we cannot simply guess which sample would be the best conducting one and further IS investigations are essential to answer the question. The suggestion to perform IS investigations is valuable and we are going to do it. Nevertheless, in the light of the obtained results, we decided that this paper will be fully devoted to the structural investigations.

Round 2

Reviewer 3 Report

The revised version can be published in MDPI. The inquiries from referees are considered and answered thoroughly.